# Moses Mendelssohn as an Influence on Hermann Cohen's "Idiosyncratic" Reading of Maimonides' Ethics

## George Y. Kohler

Department for Jewish Philosophy, Bar Ilan Universiy, Ramat Gan 5290002, Israel; george-yaakov.kohler@biu.ac.il

**Abstract:** Surprisingly, there are at least three major theological subjects where Hermann Cohen seems to agree with Mendelssohn—against standard Jewish Reform theology. Even more interesting: All three points stand in connection with the religious thought of Moses Maimonides (1137–1204), the medieval halakhist and philosopher, whose radical theological ideas Mendelssohn mostly rejected and Cohen generally adopted. Should this observation be true, however, we might assume that Cohen took at least a few hints for his own reading of Maimonides from Mendelssohn. This conclusion would then in itself be surprising, because Cohen, contrary to the Jewish Reform theologians of the 19th century, and in fact contrary to everyone else, read Maimonides in what was generally called an "idiosyncratic" way: For Cohen, Maimonides was a proto-idealist, who often followed Plato much more than Aristotle, and who sometimes even anticipated Immanuel Kant. Even more exceptionally, for Cohen Maimonides' philosophy in the Guide of the Perplexed was focused on a theology of ethics rather than on a metaphysics of knowledge of the divine. I will attempt to provide proof-texts showing that on these three points Mendelssohn and Cohen are essentially in harmony. Still, my proofs for a probable Mendelssohnian influence on Cohen depend on a very close reading of both Mendelssohn's relevant passages, as well as of the corresponding texts in Maimonides.

**Keywords:** Moses Mendelssohn; Maimonides; Hermann Cohen

## 1. Introduction

Hermann Cohen (1842–1918), arguably the most important Jewish philosopher at the turn of the 20th century, was a fervent critic of the religious philosophy of Moses Mendelssohn (1729–1783). As with so many other issues of modern Jewish thought, Cohen might be seen here as the climax of a whole century of Reform theology—as summarizing and philosophizing the ideas of his predecessors during the 19th century—much more than anticipating 20th-century Jewish thought, with its return to Jewish particularism, nationalism and existentialism. Indeed, Cohen's criticism of Mendelssohn is closer to the extensive writings that, for example, Rabbis Samuel Holdheim (1806–1860) and David Einhorn (1809–1879) devoted to the thought of the Berlin Enlightener than to the positive adoption of some Mendelssohn's teachings by Franz Rosenzweig.[1] Like these radical Jewish Reform theologians, Cohen opposed first and foremost Mendelssohn's famous definition of Judaism as "revealed law" as opposed to "revealed religion". In at least two important places in the Jewish writings of his last years, Cohen offers sharp criticism of Mendelssohn's philosophy of Judaism.[2] Mendelssohn's rejection of religious doctrine (i.e., of Jewish dogma) in favor of religious deeds (or abstaining from deeds, as in the dietary strictures or Sabbath laws) did not take into account that the acceptance of dogma is not necessarily blind, as Mendelssohn assumed, thus keeping "reason in chains" (Mendelssohn 2011, p. 124). To the contrary, Cohen claimed, *Torah* does not mean Law but doctrine, teaching—it is religious knowledge (*Erkenntnis*), and therefore (rational) dogma is actually "what bestows truth upon religious law in the first place" (Cohen [1915] 1924b, p. 259).[3]

As just mentioned, this critique of Mendelssohn was widespread in German Jewish thought during the 19th century. Surprisingly, however, there are at least three theological

---

topics about which Cohen seems to take Mendelssohn's side, sometimes against standard Jewish Reform positions. While I would not go as far as Gideon Freudenthal, who wrote in 2012 that Cohen's "own philosophy of Judaism is clearly an elaboration of Mendelssohn's ideas" (Freudenthal 2012, p. 10),[4] there are similarities and there is continuation, as this essay will attempt to show. Even more interesting: all three points of agreement stand in connection with the religious thought of Moses Maimonides (1137–1204), the medieval halakhist and philosopher, whose radical religious ideas Mendelssohn frequently rejected and Cohen generally adopted—although Cohen approved of Maimonides mostly only after a process of 'regulative idealization' in Neokantian terms.[5] Should my observation prove true, however, and there are indeed points of agreement, we might assume that Cohen took a few hints for his appropriation of Maimonides from his reading of Mendelssohn.[6] This conclusion would then in itself be surprising, because Cohen, contrary to the Jewish theologians of the 19th century, and indeed contrary to everyone else, read Maimonides in what was has been called an "idiosyncratic" way (See, for example, Hughes 2010; Seeskin 2012). For Cohen, Maimonides was a proto-idealist, who, against his own declared preferences, often followed Plato much more than Aristotle, and who sometimes even anticipated Immanuel Kant.[7] Even more exceptionally, for Cohen, Maimonides' philosophy in the *Guide of the Perplexed* was focused on a theology of ethics rather than on a metaphysics of knowledge of the divine.[8] Notably, my arguments for a probable Mendelssohnian influence on Cohen depend on a very close reading of both the relevant passages in Mendelssohn and the corresponding texts in Maimonides. I will suggest that Cohen agrees with Mendelssohn on the following points, all of which are elements of what might be called a religious ethics:

1.　All commandments (and not only some of them) are in fact מצוות בין אדם וחברו. (i.e., inter-human, ethical duties). The category of duties towards God (alone) does not, and actually cannot, exist.
2.　In this sense, all commandments that are rituals and seem to lack a rational basis are instituted in order to oppose and eventually eradicate idolatry. That is what ultimately turns ritual law into rational, moral law, because it is now directed at an end, which then can be translated into *Tikkun Olam* (improving the world), and therefore
3.　Judaism teaches that moral *good and bad* are not socially based categories, that is, general agreements within cultures and societies (in Maimonidean terms, *mefursamot*), but axiomatic, reason-based principles (*muskalot*).

　　I will attempt to provide proof texts showing that on these three points Mendelssohn and Cohen are essentially in harmony, and that both thinkers are thus continuing (with or without directly referring to him) Maimonides' teachings concerning the first two issues, and jointly and explicitly rejecting Maimonides' view on the third one.

## 2. God against Man

　　Jewish religious law, for Cohen, is a developing, autonomous ethical legislation. It is, in fact, the "Jewish" positive-historical specification of Kant's Categorical Imperative. In his final work from 1918, Cohen frequently insisted that the traditional law of Judaism is essentially moral law (*Sittengesetz*), or at least an educational means (*Erziehungsmittel*) for moral law (Cohen [1918] 1929, pp. 401, 399). He offered several philosophical justifications for this claim, derived from his concept of God. In a nutshell, the deduction runs like this: in rational thought, commandments, contrary to laws of nature, do not have reasons (*Ursachen*), but ends (*Zwecke*), like all positive law. Thus, commandments are never an end in themselves, but exist for a knowable purpose. As an object of knowledge, however, "the commandment can only then be established as a divine commandment, if God is the God of holiness or of morality". The ultimate justification for all commandments is therefore divine morality and thus all Jewish legal regulation must be evaluated accordingly: can it be an "appropriate means to serve this exclusive purpose"? (Ibid., p. 412). In this sense, Cohen argued, Biblical commandments that concern interhuman relations are a subgroup

of the commandments concerning the correlation between man and the moral God, and certainly not a separate category (Ibid., p. 405). It is at this point that we might detect the first interesting similarity with Moses Maimonides, which is later also reflected in the thought of Moses Mendelssohn.

As is well known, Maimonides rejects the division of the Biblical commandments into those knowable by the intellect and those knowable by revelation alone, a division that was introduced into Jewish religious philosophy in the ninth century by Saadia Gaon and can be found in varying forms in the writings of all of Maimonides' medieval Jewish predecessors.[9] In his *Guide of the Perplexed*, Maimonides has harsh words for those who believe that the proof for the divine origin of any given commandment is that it lacks purpose and must thus be followed blindly. In his view, this proof even amounts to blasphemy. For him, the opposite is true: all of the 613 commandments of the Torah are given by God in His wisdom for one of three basic reasons—knowledge, morality, or the practical governance of the state.[10] By logical derivation from the necessary assumption of God's infinite wisdom, we arrive at the conclusion, according to Maimonides, that even if we fail to understand the rationale of a specific commandment, this does not mean that it is irrational—we have simply not yet grasped its reason. Therefore, in the *Guide*, Maimonides proposed first to return to, but then to completely re-interpret, the traditional, Mishnaic division of the commandments, that is, the division into those between man and God on the one hand, and those between man and his fellow human being on the other.[11]

Towards the end of chapter III: 35 of the *Guide*, Maimonides sorts all the 14 content-based categories of Biblical commandments into one of these two main groups. Only very few are placed into the "between man and his fellow man" group; the vast majority are placed in the "between man and God" group. At this point, a sentence appears in the *Guide* whose understanding is decisive for my argument. Maimonides writes:

> The purpose of every commandment, whether it is a prescription or a prohibition, is to bring about the achievement of a moral quality or of an opinion [. . . ], which concerns the individual itself and its becoming more perfect; therefore it is called by them [the Talmudic sages] 'between man and God', even though in reality it affects relations between man and his fellow man.[12]

There are two ways to read this statement. The first is that it is referring to all the commandments in general, while the second is that it is characterizing the commandments between man and God in particular.[13] It could either mean, therefore, that *all commandments* (both groups) are in a certain way 'between man and God', or that the group called 'between man and his fellow' is independent, though some commandments between man and God also involve actions vis à vis one's fellow man.[14]

Hermann Cohen clearly read the statement the first way, saying that Maimonides aimed to eliminate the distinction between the categories. Cohen is thus able to ground his own view on this Maimonidean teaching, the view that all divine commandments are but a means to an end—and that this end is the ethical imitation of God as the highest moral principle (Cohen [1918] 1929, p. 414). The other and more commonly accepted possibility, that this statement is referring only to the group of commandments between man and God, is probably preferable from a linguistic perspective, but even so, Cohen is able to make some serious arguments for his own reading of Maimonides on this very point. In rabbinic literature, the notion that criminal acts between humans must be considered as a transgression against God Himself is dominant indeed. Not only is murder interpreted as a grave sin against God as the creator of all life,[15] but in a famous Talmudic passage, the sages bring in God in order to explain the ruling that theft under the cover of darkness is punished more severely than armed robbery during daylight, for the furtive thief is said to believe that "the Eye from above" does not see him, which is considered worse than the crime itself.[16]

Interestingly now, Cohen also has a predecessor and thus a strong ally in Moses Mendelssohn for his interpretation of Maimonides' statement as eliminating the distinction

between the two groups of commandments, namely, those between man and God and those between fellow men. In two different places in his *Jerusalem*, Mendelssohn takes a tough stance against the division of the commandments into these groups. Mendelssohn even blames the evil that is purported to be 'coming from religion' on the basic misconception that man has duties towards God that are independent of his duties towards humanity. Thus, Mendelssohn writes:

> God is not a being who needs our benevolence, requires our assistance, or claims any of our rights for his own use, or whose rights can ever clash or be confused with ours. These erroneous notions must have resulted from the, in many respects, inconvenient division of duties into those toward God and those toward man. The parallel has been drawn too far. Toward God—toward man—one thought. Just as from a sense of duty toward our neighbor we sacrifice and relinquish something of our own, so we should do likewise from a sense of duty toward God. Men require service; so does God. The duty toward myself may come into conflict and collision with the duty toward my neighbor; likewise, the duty toward myself may clash with the duty toward God (Mendelssohn 1983, p. 57).

However, this is not merely a theoretical error in Mendelssohn's view, or an issue of misguided theology. Rather, he claims that:

> all the evils which from time immemorial have been perpetrated under the cloak of religion by its fiercest enemies, hypocrisy and misanthropy, are purely and simply the fruits of this pitiful sophistry, of an illusory conflict between God and man, the rights of the Deity and the rights of men (Ibid., p. 58).[17]

Clearly, we are witnessing an ethical argument, leading from Maimonides to Mendelssohn and on to Cohen, if Cohen's exceptional reading of Maimonides' passage is correct. This is despite the fact that Cohen does not refer to Mendelssohn here, although he knew his *Jerusalem* well, nor does Mendelssohn refer to Maimonides. It is possible that Mendelssohn was (mis)led by the medieval (Ibn Tibbon) translation of the *Guide* that he used, or he did not presume in the first place that Maimonides held such a radically moral opinion as he held himself —that the division of the commandment categories was not only *erroneous* but outright dangerous for human society. Cohen's silence regarding his ally Mendelssohn is more suspicious and not easily explained. The answer can probably be found in the conspicuous inversion of the argument in Mendelssohn: while for Maimonides and Cohen inter-human relations are a subgroup of divine-human relations, for Mendelssohn one may not serve God if it means hurting his fellow man, apparently making the service of God a part of moral inter-human relations. This difference emerges because (at least for Cohen) God is the rational, a priori idea of truth (and also of the truth of ethics), while for Mendelssohn God remained in some ways a metaphysical substance that reason had to protect from practical misinterpretations. Ironically, immediately following his aforementioned discussion of *Guide* III: 35, Cohen criticizes Mendelssohn once again for his separation of legal Judaism from human reason (Cohen [1918] 1929, p. 415).

### 3. Law against Idolatry

From the mid-nineteenth century on, the majority of the German-Jewish Reform theologians suggested abandoning many elements of the non-rational part of Jewish religious law —not necessarily because they were irrational and ritualistic, as this could still have met their search for a new spirituality of Judaism—but because they erected artificial social barriers between the Jews and their German fellow citizens, as those rabbis argued.[18] Unlike the ancient (and probably still the medieval) non-Jews, against whom those ritual separation laws were aimed, modern-day Germans had not only ceased to persecute and ghettoize their Jewish neighbors, they were also no longer to be considered idolaters. Cohen disagreed here with his predecessors.[19] Interestingly, the disagreement concerns less the analysis—Cohen, too, believed that Jewish ritual law has first and foremost nationalistic signification; it was a "national ferment" (Cohen [1918] 1929, p. 417). The *knowable purpose*

of the ritual commandments was the national isolation of Judaism, which Cohen saw first and foremost as a religious isolation and not merely a social one, as it was for the Reformers. Religious isolation was in the past indispensable for the emergence and protection of the idea of pure monotheism, emphatically also in order "to preserve the undiminished value of Jewish monotheistic belief vis à vis the two other forms of monotheism" (Ibid., p. 418). Where Cohen parted ways with the mainstream of nineteenth-century German-Jewish theology is the question of the *future* of this cultural isolation of Judaism, about its continued necessity for the survival of monotheism in its Jewish manifestation. Cohen unambiguously confirmed this necessity: as long as Jewish monotheism is not "devaluated, replaced, or replaceable" (Ibid., p. 422) by the other forms of monotheism, its continuity is necessarily bound to the *isolation* of the Jews, not as an independent nation, but as a religious nationality within a nation.[20] It seems that for Cohen, Jewish ritual law remains a bulwark against the less pure forms of monotheistic religion, protecting the religious identity of the Jewish people, and thus the very essence of Judaism itself. Only when an ideal *religion of reason* is achieved by humanity, that is, when Judaism's form of monotheism is universally accepted and practiced, can this bulwark fall.[21]

In a similar vein, Maimonides had argued at length that Biblical law is a defense against idolatry, at least concerning those parts of it that would otherwise have no rational justification. Many mysterious commandments, he claimed, can be understood through study of the idolatrous customs of the ancient Sabians[22] —under whose influence the Biblical patriarch Abraham grew up, as Maimonides believed.[23] In Maimonides' history of idolatry, Abraham rediscovered and spread monotheism in the midst of the Sabian civilization.[24] Mosaic Law came to continue Abraham's mission, and one of its main purposes is to eradicate idolatry. As we saw above, all commandments have a purpose, according to Maimonides. Therefore, he insists, we need to study the old Sabian books if we want to understand the ritual commandments of the Torah that were given for the purpose of the fight against idolatrous beliefs:

> The knowledge of these theories and practices is of great importance in explaining the reasons of the commandments. For it is the principal object of the Law and the axis round which it turns, to blot out these opinions from man's heart and make the existence of idolatry impossible.[25]

The idolatrous practices described in those Sabian works should be eradicated from Israel by fulfilling commandments that otherwise have no rational explanation, like the prohibition of mixing linen and wool.[26] This commandment, Maimonides assumes, was given because the pagan priests wore vestments made exactly as specifically prohibited by the Torah in the given example, or because of other forms of pagan dress, custom or worship.[27] Prohibiting Israel from wearing such garments, or from performing such customs, Maimonides seems to conclude here, would thus protect the exclusively Jewish belief in the One God by erecting a bulwark of contrary cultural practice.

As far as it goes, the *Guide*'s explanation of ritual law is historical rather than functional, and leaves open the question of why, for Maimonides, this law must still be kept in his day. However, it is perhaps rather the enforcement of pure monotheism, as posited by Maimonides, which motivates Cohen's position, since the belief in a unique God is ultimately the crucial aspect of the Jewish religion for both thinkers. Universal morality can only be achieved—this is Cohen's new and central insight—if a purely monotheistic religion is guaranteed: ethics is intrinsically connected to monotheism; idolatry necessarily leads to immoral behavior. Maimonides understood Jewish law not only as the very institution that first overcame polytheism, Cohen emphasized; in the *Guide*, Mosaic Law is also construed as an effective prophylaxis against the reappearance of non-monotheistic views in whatever form (Cohen [1918] 1929, p. 395).

Interestingly, at this point Cohen is again closer to Moses Mendelssohn's conception of the fate and future of the Jewish ritual regulations than to that of the early Reform Movement. As mentioned above, Cohen developed many of the theological ideas of

the German Reformers into a philosophical, neo-Kantian climax at the beginning of the 20th century. These ideas included ethical monotheism, a transformed view of Jewish messianism, and a substantial rejection of Mendelssohn's strict separation of particular, historic Judaism from universal human reason and its salvational powers. However, regarding the question of the continued separating function of Jewish religious law, it seems that Cohen suddenly switched sides. This insistence on the further strict abidance of law seems to be only a logical, protective consequence of their common rejection of Christian theology. Nonetheless, many liberal German-Jewish theologians were ready to sacrifice ritual law, at least to a certain extent, in order to avoid the "cultural isolation" (as Cohen called it) of German Jewry.[28]

Yet, this surprising agreement between Cohen and Mendelssohn on ritual law, too, must be seen in the light of a specific reading of the post-*Jerusalem* Mendelssohn, probably even in contradiction to what he wrote before, in his main work. In *Jerusalem*, Mendelssohn largely refrains from providing functional reasons for the biblical commandments. Only in one place does he explain that all Mosaic regulations "refer to, or are based upon, eternal truths of reason, or remind us of them, and rouse us to ponder them" (Mendelssohn 1983, p. 99). Still, this concession does not seem to be Mendelssohn's justification for *keeping* those regulations, or for lending them authority. The remark sounds rather like a mindful game of the law-observing philosopher building tentative intellectual bridges—because, in order to be *reminded* of an 'eternal truth' by a mitzva, you have to know the truth first. In a later passage of *Jerusalem*, Mendelssohn indeed made a connection between idolatry and ceremonial Jewish law. "Images and hieroglyphics lead to superstition and idolatry, and our alphabetical script makes man too speculative", Mendelssohn argued there; in order to circumvent those dangers, "the lawgiver of this nation gave the ceremonial law". Due to its ritual character this law consists only of actions, and "man's actions are transitory; there is nothing lasting, nothing enduring about them that, like hieroglyphic script, could lead to idolatry"(Ibid., pp. 118–19).[29] The connection made here between ritual commandments and the prevention of idolatry came thus first and foremost to protect the Jewish people itself from superstitious practice, and the remedy is not to be found in the given commandment, but rather in the transitory nature of a human deed in general, because our actions are not prone to be understood as divine in themselves. In *Jerusalem*, thus, Mendelssohn ultimately leaves the reader with the idea that Mosaic law must be kept because of the 'historical truth' of the event at Mount Sinai, and because of the authority lent to this revelation by the many Israelites bearing witness to it.

It was only after he published *Jerusalem*, in a letter to Herz Homberg from 1783, that Mendelssohn "provides some important clarifications of his views" on the ceremonial law, as Michah Gottlieb commented (Mendelssohn 2011, p. 71). Homberg, who had taught Mendelssohn's children and worked with him on the commentary to his Pentateuch edition, apparently asked Mendelssohn immediately following the publication of *Jerusalem* to further explain his insistence on the continued validity of Jewish ritual law—indicating strong disagreement of his own. It seems that Homberg referred in his (now lost) question to the very passage regarding hieroglyphic script quoted above, and it might be further speculated that he had noted what some of Mendelssohn's later critics in the 19th century pointed out: even if the ceremonial action law was the perfect sign language to avert idolatry, and to induce men to engage in reflections about the eternal truths, it should by now have long achieved this intended result and have cured the Jews of superstition. Therefore, ceremonial law should in fact have abolished *itself*.[30] In his answer to Homberg, Mendelssohn wrote that indeed he is of a different opinion regarding ritual law. Even if its significance as a sign language may have lost its usefulness, its necessity "as a unifying bond" has not yet come to an end; to the contrary,

> this bond will have to be preserved by the plan of providence as long as poly-theism, anthropomorphism and religious usurpation dominate the world. As long as these tormentors of reason are unified, must [not] genuine theists also

form some kind of union [*Verbindung*] if these [tormentors] are not to trample everything under foot? (Mendelssohn 2011, p. 124[31])

The statute of this theistic union, Mendelssohn explained to Homberg, would be the prescribed actions of the ceremonial law, because, alternatively, doctrines of faith, symbols and formulas would "keep reason in chains", as do idolatry and superstition (Ibid.).[32] The letter is especially interesting because Mendelssohn was, at least outwardly, an ardent proponent of religious tolerance.[33] Unlike Maimonides and Cohen, he did not maintain the same definition of idolatry for Jews and non-Jews, but would allow for combined forms of worship as long as the one, true God is part of them—following his view of the historical truth of the forms and rituals of worship, which was opposed to the philosophical approach to this question held by Maimonides and Cohen.[34]

However, who are Mendelssohn's genuine theists? In an important 1929 article on Mendelssohn's concept of religion, Fritz Bamberger has shown that for Mendelssohn there can be actually no true theists but Jews. For non-Jewish theists to keep the ceremonial law of Judaism, as the statues of the theistic union demanded, would stand in strict contradiction to all that Mendelssohn ever wrote about the particular nature of that law. Therefore, here we have his new and probably most functional definition of the purpose of Jewish ritual—which agrees, to a large extent, with what Maimonides and Cohen wrote about the same subject: it is a bulwark against the idolatrous beliefs of the outside world, intended to preserve the purity of Jewish faith. One could even say: it is meant to preserve Jewish theism and not necessarily Jewish theists, who are only the means to this end. In this letter, Mendelssohn felt the urgent need to call for the establishment of a theist union, which is not given by the very existence of the Jewish people. For Mendelssohn, as later for Cohen, it now seems, "the Jewish people exist to serve Judaism, Judaism is not the means of uniting people who happen to share an ethnicity", as Robert Erlewine has put it (Erlewine 2015, p. 314). Contrary to Maimonides' view of Jewish ritual law, for Cohen and for the Mendelssohn of the Homberg letter, the actual content of the ritual has become unimportant and can thus be attributed to tradition. What remains significant is only the *collective performance* or avoidance of the action; this is what creates an internal bond and outward isolation, no matter if this action is eschewing pork or laying tefillin. Still, the mitzvot are not interchangeable with other actions to achieve the same goal, but only because they were Jewish custom for thousands of years and have thus established identity. Even Cohen seems to believe that none of the great universal values brought by Judaism to human civilization was as likely to keep the Jews together as eating matzah on Pesach.

## 4. Ethics against *Consensus Gentium*

The third and last point of agreement between Cohen and Mendelssohn concerns the religious connotations of the a priori character of ethics, that is, the belief in the human ability to arrive, by pure thought, at universal, timeless laws determining what is good or evil. This is not entirely unimportant in terms of religion, because the possibility of a priori ethical knowledge warrants the idea of autonomy, of moral *self-legislation* of the acting human being. For Cohen, this ability is almost self-evident; he is the author of a systematic, neo-Kantian philosophical ethics, seen by many as the climax of his three-volume *System of Philosophy*, the *Ethik des reinen Willens*, first published in 1904. In Kantian ethics, in order to pass the test of the in-itself purely formalistic Categorical Imperative, moral axioms have to be derived by reason a priori, with no regress to the empirical world. Being more radical than Kant himself, Cohen saw a mutual and intrinsic connection between ethics and logic in his reinterpreted Kantian concept of the *noumenon* as an ideal, rational construct. It is indispensable for the progress of science but, on the other hand, clearly an ethical concept in itself, an "ought". For Cohen, the noumenon is not a given, but an infinite task; it constitutes the "ought" of the scientific ideal of the perfect, mathematical, pure constitution of its empirical object. The noumenon can thus, as *pars pro toto*, stand for both the ethics of science and for the scientificity of ethics.[35] In addition, Cohen's very concept of God is derived from the concept of a priori truth, that is, of "the necessary connection of the

knowledge [*Erkenntnis*] of nature and the knowledge of morality". More specifically, for Cohen, 'God' means the accordance [*Übereinstimmung*], that is, the systematic unity, "of theoretical causality and ethical teleology" (Cohen [1918] 1929, p. 476).

Given this theoretical and rational foundation of morality, Cohen would certainly include ethical truth in what Maimonides called *muskalot*, the demonstrable axioms of reason. In contrast, for the medieval philosopher himself, moral behavior evidently belonged to the category of *mefursamot*, generally accepted views or knowledge, neither belonging to tradition nor to reason but to the sphere of ever-changing social agreements that humans enter into for the sake of a harmonious society. In this sense, for Maimonides, ethics would have no place in either speculative philosophy or authoritative religion—a position that goes directly back to Aristotle. In 1908, Cohen devoted an almost book-length essay to the question of why Maimonides was drawn into the 'trap of Aristotle' on this subject; why he believed, that is, that ethics is determined by a 'golden middle path' between extremes, and that ethical laws amount, in fact, only to a relativist *consensus gentium* in a given time period, area or culture. Interestingly, this reading of Maimonides' general ethical approach is not even undisputed. Although it seems to be beyond doubt that the young Maimonides, in a work known today as the *Eight Chapters,* presented clear-cut Aristotelian relativism, and that this view is repeated, more or less, in his monumental halachic work—at the far other end of Maimonides' life's oeuvre, in the very last chapter of his last work, the *Guide of the Perplexed*, appears at least a hint to ethical primacy. In what Steven Schwarzschild (following Julius Guttmann) called an "ethical twist", Maimonides here suddenly declares that the final purpose of knowing God is not a theoretical comprehension of His essence but the appreciation of His ethical action as a model for human imitation.[36]

Cohen, however, would not deny the Aristotelian influence on Maimonides in questions of morality. He thinks it understandable, if not forgivable, that, by placing morality under the term *mefursamot*, Maimonides "underestimated the danger of Aristotle's depreciation of ethics" (Cohen [1908] 1924c, p. 244)—because the only reason Maimonides did not recognize this problem is that he understood ethics as an inherent part of Judaism. If for Maimonides intellectual perfection is the highest goal of religion, and the result of this perfection is true knowledge of God, then ethics as rational knowledge is secured in religion, Cohen argued.[37]

Interestingly, Moses Mendelssohn, too, seems to have thought a great deal about the a priori nature of good and evil in connection with Maimonides. It might even be that Maimonides' relativist Aristotelianism on this point provoked Mendelssohn into stating his own, opposite position several times in the Maimonidean language of *muskalot* and *mefursamot*. Mendelssohn is clearly convinced that ethics belong to the *muskalot*, and is thus engaged in a mighty, lifelong struggle with the view of the great Maimonides, which he could not comprehend, as he explicitly stated.

In 1763, Mendelssohn won a prestigious prize from the Royal Prussian Academy with an essay on the question of whether metaphysical truths in general, and moral truths in particular, are susceptible to the same evidence as mathematical truth. His answer was a hesitant affirmation.[38] Mendelssohn argued in his prize essay (in the wake of Christian Wolff) that although there is something like a "moral sentiment" in human beings, it would not represent the principle of morality, which can only be established by a clear and rational understanding of good and evil (Altmann 1973, p. 126). During the same years, Mendelssohn wrote a Hebrew commentary on Maimonides' early treatise on *Logical Terms*.[39] Discussing Maimonides' category *mefursamot* (general social agreements), Mendelssohn in his commentary repeated the view from the *prize essay*: the human heart feels pleasantness if confronted with justice and lawfulness and pain if it encounters violence, quarrel and human oppression "even before reason investigates good and evil (טרם חוקר השכל אחרי הטוב והרע)".[40] While those feelings obviously belong to the *mefursamot*, here discussed, there is little doubt that, although feelings are first affected, the investigation of morality *by reason* is what Mendelssohn saw not only as possible and fruitful, but also as excluded from the *mefursamot*.[41]

Even more explicit in this respect is a passage from a letter Mendelssohn wrote in 1773 to Rabbi Yaakov Emden. Three years earlier, in his famous dispute with Johann Caspar Lavater, Mendelssohn had claimed that, unlike Christianity, the Jewish religion did not need to condition salvation on the joining of its ranks. For the purpose of granting non-Jews a place in the 'world to come', the sages of the Talmud had instituted seven "Noachide Laws" that, when kept, guaranteed eternal bliss.[42] In his letter to Lavater, Mendelssohn went one significant step further and equated those seven precepts with the 'natural law' of human reason. In an obvious attempt to include also those cultured gentile peoples, newly discovered during the past few centuries, who had no access to the Bible, he thus implicitly declared morality again to be a priori ('natural') knowledge.[43] However, the legal authority of Maimonides stood in the way of this conflation of Noachide and Natural Law. In a controversial statement Maimonides ruled that, to provide salvation, these seven precepts must be accepted by the people who follow the Noachide law as part of the teachings of Moses, and not only on the basis of their inherent rationality.[44] Writing to Lavater, Mendelssohn simply dismissed Maimonides' precondition on the basis of it not having Talmudic precedent.[45]

In his letter to Rabbi Emden from three years later, Mendelssohn still seems to be struggling with the problem posed by Maimonides, surprisingly sounding quite desperate. One way to solve the problem, he suggested, is to refer to Maimonides's general negative view of the rationality of morals. Now writing in Hebrew, Mendelssohn complains openly that in his eyes, Maimonides' "opinion of our knowledge of good and evil, and his excluding it from the *muskalot* is very strange (זר מאוד)". He himself, Mendelssohn continues, possessed "clear and proven evidence for how good and evil, justice and wrong are bound in reality to reason" (Mendelssohn 1972, p. 179; my translation). Additionally, if that outspoken critique of Maimonides was not clear enough, in his Hebrew commentary (*Bi'ur*) on Genesis 2:9 (the *Tree of Knowledge of Good and Evil*), written shortly afterwards, Mendelssohn continued the same reproach. This verse is of special interest because Maimonides also used the Eden metaphor in a central text of his discussion of the *mefursamot* character of moral judgments. In chapter I,2 of the *Guide*, Maimonides explains that only truth and falsity are pure intellectual concepts, while our notions of beauty and ugliness belong to generally agreed-upon aesthetics.[46] They are not described as true or false, but rather as good or bad.[47] In his own interpretation of the *Tree of Knowledge*, Mendelssohn repeats that he would not agree with Maimonides concerning the matter of the *mefursamot*, and continues thus:

> It seems that in fact also good and evil belong to the *muskalot* [reason based principles]. Pleasant and obscene are not synonymous terms for the good and the evil. This is because good and evil are determined by reason and pleasant and obscene are determined by the senses (Mendelssohn 1990a, 2:22–23). [48]

This claim of his is easily demonstrated, Mendelssohn continued, but here (in a Torah commentary) was not the place to go into detailed philosophical proof. He had already provided his evidence for the rationality of ethics in the prize-winning essay from 1763.[49] In his commentary on Genesis, as in the other places discussed above, the notable feature is Mendelssohn's declared opposition to a giant in Jewish thought, which he in no way tried to explain away or to harmonize with his own clearly antithetical view.[50]

On this last point, the existence of a priori philosophical ethics, Hermann Cohen was in no need of taking hints from Moses Mendelssohn. It might be possible, however, that Cohen was motivated by Mendelssohn to rethink Maimonides' strict Aristotelism in ethical questions, and to try to find more sophisticated explanations for this striking difference between Mendelssohn's, his own, and Maimonides' position. However, even this is rather unlikely. What probably can be said is that Cohen's final word on Mendelssohn's supposedly erroneous and harmful theory of Jewish law was less harsh, less condemnatory and even less final because of the three points discussed in this paper—subjects concerning which we find surprising agreement between Cohen and Mendelssohn, despite Cohen's overall scathing critique of the legal philosophy of Mendelssohn's *Jerusalem*. Towards the

very end of his last book, the *Religion of Reason*, Cohen writes that if what we all believe Mendelssohn to have claimed in *Jerusalem* was true, he would either have been a hypocrite, a doctrinaire or simply short-sighted. However, saying so is to commit a grave injustice "against the memory of this new Moses". Therefore, Cohen concludes, Mendelssohn's position regarding the Law is yet in need of a "better illumination", obviously including Cohen's own view of it (Cohen [1918] 1929, p. 416).

**Funding:** This research received no external funding.

**Conflicts of Interest:** The author declares no conflict of interest.

## Notes

[1]　On Einhorn and Mendelssohn, see (Greenberg 1982). On Holdheim's critique, see (Kohler 2020).

[2]　(Cohen [1915] 1924b; Cohen [1918] 1929, pp. 415–16). For an extensive and insightful discussion of those texts, see (Erlewine 2015).

[3]　My translation. For Mendelssohn, in contrast, the authority of the religious law of Judaism was rather a consequence of a historical truth, of the Israelites bearing witness to and handing down the tradition of the revelation at Mount Sinai.

[4]　In a footnote, Freudenthal qualified his claim as referring basically to Cohen's "view of the law", which "perfectly agrees with Mendelssohn's". The difference is only in the "theoretical foundation"; Cohen would reach "some of Mendelssohn's conclusions only after various meanders" (Ibid., p. 429).

[5]　For a discussion of this method in Cohen, see (Schwarzschild [1979] 2018a).

[6]　Openly, however, Cohen accused Mendelssohn of having "left the path of Maimonides, and thus the entire doctrine of Jewish faith" (Cohen [1915] 1924b, p. 259). Mendelssohn's above-mentioned theory of *bearing witness* seems indeed rather adopted from Yehuda Halevi's *Kuzari* (11th century) and is not found in Maimonides.

[7]　See Cohen's essay on Kant and Jewish philosophy, (Cohen [1910] 1924a).

[8]　For Cohen's reading, see first and foremost (Cohen [1908] 1924c); for an English translation, see (Cohen 2004).

[9]　The distinction itself is first found in the Sifra on Lev 18, 4 (140) as the distinction between *mishpatim* and *chukkim*.

[10]　See Maimonides' *Guide* III: 31.

[11]　M Yoma 8:9. The context in the Mishnah is actually the difference in the conditions for atonement for sins/transgressions belonging to the respective groups. To designate God, the Mishnah uses the word *hamakom*, meaning "the Place".

[12]　My own translation.

[13]　The original Arabic phrase seems to be ambiguous. Both Ibn Tibbon and Kapach render the Hebrew translation as follows:

ושאר המצוות הם בין אדם למקום. והוא שכול מצווה . . .

To introduce this sentence with "והוא" clearly means reading it the second way. The Hebrew translation by Michael Schwarz, deviating from all former translations into Hebrew, (probably unknowingly) supports Cohen's reading of the passage (Schwarz 2002, p. 556). In the German translation published in 1923 by A. Weiss (after Cohen's death) it reads, "Man nennt nämlich jedes Gebot . . . " and Weiss adds a note in favor of Cohen's reading (see Maimonides 1923–1924, p. 218, note 25). Pines and Munk write "For every commandment . . . " and "Par tout commandement . . . ", respectively, which leaves it more open.

[14]　This is the way Hannah Kasher reads the Maimonidean definition (probably she used the translation by Ibn Tibbon). See (Kasher 1984, pp. 23–28).

[15]　R. Akiva in Tosefta Yevamot 8:5. In his *Laws of Murder and Saving Souls* (1:4), Maimonides writes that the soul of the murdered is the personal property of God and cannot be ransomed.

[16]　TB Baba Qama 79b. To complicate matters, this Talmudic explanation is one of the few instances where Maimonides in the *Guide* does not adopt the reason the sages gave for the ruling (see *Guide* III: 41). However, this does not necessarily mean that he disagreed with their explanation.

[17]　See here Maimonides' Laws of Shabbat (2.3): It is even forbidden "to *hesitate* before transgressing the Sabbath [laws] on behalf of a person who is dangerously ill. [ . . . ] Concerning those non-believers (*epikorsim*) who say that [administering treatment] constitutes a violation of the Sabbath and is forbidden, one may apply the verse [Ezekiel 20:25]: *I gave them harmful laws and judgments through which they cannot live . . . "* Maimonides is rejecting here the same idea that not helping your fellow man could find favor in the eyes of God. The purpose of the law is to "bring mercy, kindness, and peace to the *world*", and not to curry divine favor.

[18]　This is especially obvious in the writings of Samuel Holdheim, who argued that continued cultural separation (for the sake of the law only) would infinitely delay the coming of the Messiah (see Holdheim 1845).

[19]　For this disagreement, see in detail (Kohler 2018, pp. 191–92).

[20]　For an in-depth discussion, see (Erlewine 2015, pp. 313–17), pointing out interesting differences between Cohen's two accounts of Mendelssohn's view of the law.

21    For Cohen's discussion about the talmudic passages announcing the repealing of the Law in the Messianic era, see (Cohen [1918] 1929, p. 424). Ultimately, however, Cohen's Messianism is an infinite advent, always approaching but never achieving its purpose.

22    Knowledge of the Sabians came to Europe through Maimonides' account in the *Guide*. Beginning in the seventeenth century, intensive research was done in order to uncover the history of this people that lived in Mesopotamia. However, Maimonides himself seems to have followed the Arab custom of his time in calling the religion of all idolatrous peoples *Sabian*. See on this subject (Elukin 2002).

23    *Guide* III: 29 (beginning)

24    *Mishneh Torah*, Laws of Idolatry, chp. 1.

25    *Guide* III: 29 (end).

26    *Guide* III: 37.

27    A detailed description of those Maimonidean teachings can be found in (Kreisel 2008, pp. 162–66).

28    In 1845, Abraham Geiger wrote a private letter to Leopold Zunz demanding explanations why Zunz, as rumors had it, had re-introduced a kosher kitchen into his home. Geiger explicitly states in this letter that on the one hand "the dietary laws, of all things, are so very spiritless and so very much obstruct social life" and on the other hand "the profoundest fraternization of men is more important than the revitalization of a separatist and dubious religious feeling" (Geiger 1878, p. 181).

29    This educational argument ("could lead to") is found also in Maimonides, but never in Cohen, who seems to have strong Kantian beliefs in the power of human will and agency.

30    This is what Samuel Holdheim, for example, argued against Mendelssohn in 1845, see (Holdheim 1845, p. 60; Kohler 2013, p. 193).

31    German original: (Mendelssohn 1977, p. 134). Samuel Holdheim rests much of his case against Mendelssohn on the words "as long as", arguing that they prove that even Mendelssohn would confirm the possibility of the abandonment of the ritual law of Judaism. The question is only when one would consider the *tormentors of reason* to be defeated, see (Holdheim 1845, p. 64).

32    English: Translation edited by me. The apparent similarity between Maimonides' theory of the law and Mendelssohn's was noted by some more conservative Reformers even before Cohen. See for example the short article (Biach 1905).

33    See the powerful manifesto on the last pages of *Jerusalem*. In his private letters, however, Mendelssohn is sometimes ready to claim theological superiority for Judaism, but this is not because of Christian idolatry, rather because of the absence of reason in Christian dogma and it therefore does not stand in the way of a coexistence of religions for Mendelssohn. See his letter to the Crown prince of Brunswick (Mendelssohn 1974b, pp. 303–5), but also to Elkan Herz from 1771 (Mendelssohn 1990b, p. 150, letter 127).

34    For a detailed discussion of these differences between Maimonides and Mendelssohn, see (Kaplan 1998, p. 431). For Cohen's view of Christianity, see (Lyden 1994).

35    See here (Schwarzschild 1981), p. viii. Unfortunately, in later editions of Cohen's *Werke*, this introduction was replaced by another text. It is reprinted in (Schwarzschild [1981] 2018b).

36    Referring to *Guide* III, 54, the very last paragraph of the whole work. See (Schwarzschild 1990b, p. 144); see also (Guttmann 1927, pp. 70–71). The same idea is taken over in Guttmann's major *Die Philosophie des Judentums* from 1933 (reprinted: Guttmann [1933] 2000, p. 206).

37    Cohen explained, following Maimonides (Guide I, 54), that what we know of God are only "the attributes of action" (Guide I, 52–3), the way God acts in the world. However, those are exclusively ethical attributes, as Scripture itself revealed: God is *compassionate and gracious, abounding in kindness and faithfulness* (Exodus 34:6–7). For an extensive discussion of this argument, see (Kohler 2012).

38    The essay is discussed in great detail in (Altmann 1973).

39    See (Mendelssohn 1761). In 1765, Mendelssohn published a significantly expanded version of this commentary. The Maimonidean authorship of the work is controversial. For our purposes, however, it suffices that Mendelssohn believed Maimonides wrote the treatise.

40    (Ibid., chp. 8, p. 71). English translation in (Breuer 2018, p. 84).

41    Alexander Altmann, however, wants to read this sentence as *prioritizing* moral feeling over moral thought and thus as a contradiction to the *Preisschrift*, and not only as a chronological order. See (Altmann 1969, p. 26).

42    See b. Abod. Zar. 64a.

43    For discussion, see (Novak 2011, chp. 13). See also (Ibid., chp. 14) on Cohen.

44    *Mishneh Torah*, Laws of Kings and Wars 8:11. In addition to the problematic ruling itself, most of the printed editions are corrupted at this place, further complicating matters.

45    I cannot enter here into a discussion of the interesting background of this whole issue. See the classical discussion by (Schwarzschild 1990a).

46    That nakedness is seen as ugly has no rational explanation, Maimonides argued, for example in *Guide* I, 2.

47    See here again Steven Schwarzschild with an ambitious attempt to read Maimonides as still upholding rational morality in his "Middlingness" (Schwarzschild 1990b, p. 149).

48   My translation from the Hebrew Original.

49   Elias Sacks argues concerning the two passages (from the Bi'ur and the letter to Emden) that Mendelssohn here "reflects on the intellectual distance between the twelfth and eighteenth centuries" the same way he did when discussing Maimonides's attitude toward Aristotelian astronomy. That is, Mendelssohn comes to emphasize "the importance of revising commitments in light of shifting frameworks, and the possibility that words composed at one point in time might fail to express concepts generated by later models" (Sacks 2016, pp. 84–85). I believe that, while this is probably true concerning astronomy, Mendelssohn did not believe that 'new knowledge' could arise in the field of ethics. What he argues in the *Preisschrift* (and elsewhere), that moral truth can demonstrated by reason alone, that was the case for him also in Greek antiquity. This, in turn, is the very reason why *he himself* used the word *muskalot* at both places.

50   Another essay by Mendelssohn that discussed rational knowledge of good and evil without referring to Maimonides is his late and, in his lifetime, unpublished, "Sache Gottes, oder die gerettete Vorsehung", see esp. § 70 and 83 (Mendelssohn 1974a, 2:243).

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
