# Peer review of "Moses Mendelssohn as an Influence on Hermann Cohen’s “Idiosyncratic” Reading of Maimonides’ Ethics"

_religions, doi:10.3390/rel14010065_

Round 1

Reviewer 1 Report

great article, good work!

The Schwartz Heb translation of the Guide is a bit out of place, and seems unnecessary to me. If you don't like the 2 Eng translations, why not just bring your Eng translation of the passage from the Guide? In any case, the journal's text of this Hebrew needs careful checking, it seems awry. Or ask Lenn Goodman for this forthcoming translation into English

Author Response

Corrected. I use my own translation and mention in the note that Schwarz translated the Arabic in Cohen’s sense, which I found surprising. This also deviated from all former Hebrew translations, that’s why it was important for me.

Reviewer 2 Report

This essay is an interesting exploration of the complex relations among the views of Maimonides, Mendelssohn, and Cohen. It's a complicated set of topics with many points of detail and nuance, but the author successfully navigates a way to a reasonable conclusion. The author is very explicit about the fact that his conclusions cannot be definitively proven, but fall in the category of plausibility and similarity. But the discussion is still constrictive and worthwhile in its illumination of these relations. I recommend its publication, with no revisions required.

Author Response

Thank you!

Reviewer 3 Report

well written, clear paper. i enjoyed reading this paper as it illuminated for me the philosophical connections between maimonides, cohen and mendelssohn. the paper did a good job at contrasting and comparing these three philosophers. i learned a lot! 

the only point of discomfort for me was a small discrepancy in the citations. the author chose to use hebrew for maimonidean citations when maimonides originally wrote in judeo-arabic. furthermore, when quoting mendelssohn the author sometimes does so in english (when the original would have been german or hebrew) sometimes in hebrew. a choice needs to be made: either all the citations are quoted in the original (maimonides being then quoted in judeo-arabic) or the citations are all quoted in english. i personally feel that nothing would be lost in simply quoting everything in english--it would make for smoother reading. 

Author Response

Thank you! Regarding the citations: I deleted now the Hebrew for Maimonides and use my own English. I believe the terms muskalot/mefursamot are important (and agreed upon in Maimonides scholarship), so I left them in, but only transliterated, not in Hebrew letters. I quote Mendelssohn in English where he wrote Hebrew, but put a few Hebrew phrases in brackets for the reader interested in the original formulation.